# Fermentation Quality and Microbial Community of Corn Stover or Rice Straw Silage Mixed with Soybean Curd Residue

**DOI:** 10.3390/ani12070919

**Published:** 2022-04-03

**Authors:** Xiaolin Wang, Jiamei Song, Zihan Liu, Guangning Zhang, Yonggen Zhang

**Affiliations:** College of Animal Science and Technology, Northeast Agricultural University, Harbin 150030, China; wxl547312630@163.com (X.W.); songjiamei1120123@163.com (J.S.); neau13244537135@163.com (Z.L.)

**Keywords:** soybean curd residue, corn stover, rice straw, mixed silage, fermentation quality, microbial community

## Abstract

**Simple Summary:**

Soybean curd residue (SCR) is a potential ruminant feed, offering a rich source of fiber, protein, and lipids. However, the excessively high water content of SCR may lead to difficulty in its storage. For ruminants, corn stover and rice straw are common sources of roughage, but these are often restricted because of their low digestibility. Mixed ensiling of SCR with corn stover (CS) or rice straw (RS) may provide a solution to the problem of the SCR being difficult to preserve. This study aimed to evaluate the chemical constituents, fermentation quality, and microbial community of CS or RS silage mixed with SCR. Such mixing with SCR increased the lactic acid and protein contents and decreased the pH value, the content of neutral detergent fiber (NDF) and acid detergent fiber (ADF), ammonia nitrogen concentration, and bacterial diversity in both CS and RS silage mixtures and improved their nutritional value and fermentation quality as well.

**Abstract:**

The objective of this study was to investigate the fermentation quality and microbial community of corn stover (CS) or rice straw (RS) silage mixed with soybean curd residue (SCR). In this study, SCR and CS or RS were mixed at ratios of 75:25, 70:30, and 65:35, respectively, and measured for nutrient content, fermentation indices, and bacterial diversity after 30 days of ensiling. The results showed an increase in lactic acid (LA) concentration (*p* < 0.01) and crude protein (CP) content (*p* < 0.0001), a decrease in pH value (*p* < 0.01), the content of NDF (*p* < 0.01) and ADF (*p* < 0.01), and ammonia nitrogen (AN) concentration (*p* < 0.01) as the proportion of SCR in raw materials (CS or RS) increased. The addition of SCR to silage led to a decrease in bacterial diversity and contributed to an increased relative abundance of beneficial microorganisms, such as *Lactobacillus*, and a corresponding decrease in the relative abundance of undesirable microorganisms, such as *Clostridium* and *Enterobacter*. Collectively, the mixed silage of soybean curd residue with corn stover or rice straw preserved more nutrients and helped improve fermentation quality.

## 1. Introduction

Soybean curd residue (SCR) is the main byproduct in the process of producing bean curd (tofu) and soymilk. SCR has a high nutritional value, provides a rich source of fiber, protein, and lipids, and has potential for use in ruminant feeds [1]. In addition, SCR is a rich source of bioactive compounds, such as unsaturated fatty acids, isoflavones, phenolic lignans, phytosterols, coumestans, saponins, and phytates, which not only have biological activity including antioxidant and antimicrobial properties, but could potentially contribute to the prevention of cardiovascular disease and even certain types of cancer [2]. The annual production of SCR reached 2.8 million tons in China in 2012 [1]. Such a large amount of SCR could result in serious environmental pollution if discarded improperly. However, the high water content (greater than 80%) and low sugar substrate levels in fresh SCR may make it difficult to produce high-quality silage by natural fermentation, increasing the risk of *Clostridium* fermentation and nutrient losses and resulting in excessive dry matter loss, extensive proteolysis, and an increase in butyric acid production, thereby decreasing feed digestibility, N utilization efficiency, and feed intake [1,3]. However, the problem of high-moisture content could be solved by mixing with dry agricultural byproducts such as peanut hulls or straw [4].

Corn stover (CS) and rice straw (RS) are the main residues of corn and rice production in northeast China and, therefore, are abundantly available annually [5,6]. A large amount of CS and RS are discarded or inappropriately burned each year, wasting a potential resource, and increasing environmental problems. Ensiling is a common preservation method for straw which can prolong its storage time and provide fodder for ruminants throughout the year. However, CS and RS contain low levels of crude protein and digestibility as well as prominent levels of lignin, making them poor candidates for fermentation alone which would cause rapid protein breakdown and high ammonia production, making it difficult to ensure feed quality [7,8]. However, several studies have shown that mixed ensiling could improve silage quality and promote the stability of the fermentation process compared with sole fermentation [9,10]. Mixed ensiling of SCR with CS or RS may have potential advantages: (1) CS or RS can serve as water absorbents (reducing the SCR problems of high water content and poor storage stability) and can provide additional water soluble carbohydrates (WSC) to promote advantageous silage fermentation; (2) SCR can improve the nutritional quality by increasing the protein content of fodder; and (3) the isoflavones and other major bioactive components in SCR could inhibit undesirable bacterial growth, such as *Enterobacter,* and may help with protein preservation, reduce nutrient loss, and improve fermentation quality during ensiling [11]. However, the mixed ensiling of SCR with CS or RS should be evaluated to determine the effect of different mixed ratios on fermentation quality. In addition, the microbial community related to mixed ensiling of SCR with CS or RS has rarely been investigated.

Therefore, this study aimed to investigate: (1) the nutrient content and fermentation quality of CS or RS silage mixed with SCR; and (2) the microbial community of CS or RS silage mixed with SCR. This might provide technical support for the preparation of high-quality silage and its application in ruminant feeding.

## 2. Materials and Methods

### 2.1. Raw Materials and Silage Preparation

Soybean curd residue was obtained from a plant for the processing of soy products (Harbin, China). CS and RS were cultivated at the experimental field of Northeast Agricultural University (Harbin, China). CS and RS were harvested in June 2018 and chopped to the approximate length of 2 cm using a crop chopper. Soybean curd residue (SCR) and CS or RS were mixed at ratios of 75:25 (C25/R25), 70:30 (C30/R30), and 65:35 (C35/R35), respectively. After thorough mixing, the silage mixtures for each treatment (approximately 2 kg fresh weight) were tightly packed separately in polythene bags and sealed by using a vacuum packing machine; each bag was equipped with a hole that only enabled gas release. A total of 18 bags (2 materials × 3 treatment ratios × 3 replicates) were prepared and stored at ambient temperature (25−30 °C) for 30 days of ensiling. The silage bags were unsealed to determine fermentation quality, chemical composition, and bacterial communities after the 30-day ensilage period.

### 2.2. Analysis of Microbial Population, Nutritional Composition, and Fermentation Quality

The microbial population was determined according to Ni et al. [10]; 30 g of sample was evenly mixed with 270 mL of sterilized saline, and then a series of gradient bacterial solutions were obtained by serial dilution. The lactic acid bacteria were grown at 37 °C in plate count on lactobacilli de Man, Rogosa, and Sharpe (MRS) agar medium (Sinopharm Chemical Reagent Co., Ltd., Shanghai, China), and colonies were counted 48 h later. Molds and yeasts were counted on potato dextrose agar medium (Sinopharm Chemical Reagent Co., Ltd., Shanghai, China) and then kept in an incubator at 30 °C for 2–3 days. Finally, colonies were counted as the number of viable bacteria in colony forming units (CFU) per gram of fresh matter (FM).

The dry matter (DM) content of the sample was determined by drying at 65 °C for 48 h in a forced-draft oven (DGX-9243B-1, Fuma Laboratory Co., Ltd., Shanghai, China). The dried sample was pulverized through a 1 mm screen in a grinding machine (FZ102, Taisite Instrument Co., Ltd., Tianjin, China) and analyzed for crude protein (CP) content by the methods of the Association of Official Analytical Chemists [12]. The contents of neutral detergent fiber (NDF) and acid detergent fiber (ADF) were determined following the methods detailed by Van Soest et al. [13]. Water soluble carbohydrates (WSC) were analyzed by the method described by Owens et al. [14].

In addition, the sample (20 g) was mixed uniformly with 180 mL distilled water and suspended in a refrigerator at 4 °C overnight for aqueous extraction, and then the extracts were filtered through four layers of cheesecloth. The filtrate was used for subsequent determination of the pH value, and the concentrations of organic acids and ammonia nitrogen (AN). The pH value was measured with a glass electrode pH meter (Sartorius Basic pH Meter, Göttingen, Germany). The concentrations of lactic acid (LA), acetic acid (AA), propionic acid (PA), and butyric acid (BA) were measured by high-performance liquid chromatography (HPLC) [15]. The concentration of ammonia nitrogen was determined using the indophenol-blue method [16].

### 2.3. Microbial Diversity Analysis

A 10 g sample was removed from each silage bag and 40 mL sterile saline (0.9% sodium chloride) was added and mixed thoroughly by vortexing. The filtrate was centrifuged at 10,000 r/m for 10 min at 4 °C and the supernatant was discarded. Then, the remaining sediment was suspended in 3 mL of sterile saline. Genomic DNA was extracted using the TIAN amp Bacteria DNA Kit (TIANGEN Co., Ltd., Beijing, China) following the manufacturer’s instructions. The extracted DNA was subjected to PCR using the Q5 High-Fidelity DNA Polymerase System (New England Biolabs, Ipswich, MA, USA), and the V3–V4 regions of the 16S rRNA gene were processed for amplification with the primers. The following primers were used: 338F (5′-ACTCCTRCGGGAGGCAGCAG-3′) and 806R (5′-GGACTACCVGGGTATCTAAT-3′). Purified DNA was sequenced on an Illumina MiSeq platform (Illumina, Inc., San Diego, CA, United States) at Baimaike Co., Ltd. (Beijing, China). The sequences obtained from the MiSeq platform were processed using the open-source software pipeline QIIME (version 1.8.0).

Alpha diversity indices, Beta diversity, and a bacterial composition histogram were calculated by QIIME (version 1.8.0) pipeline software. Alpha diversity indices (including the Chao1 and Shannon) were used for the richness and diversity indices of the bacterial community. Beta diversity was used to evaluate the differences in bacterial community compositions in the silage samples and visualized by principal coordinates analysis (PCoA). Furthermore, the relative abundance of the distinct bacterial communities of each silage sample was determined at the genus level, and a heatmap analysis was performed. The Venn diagram and Heatmap were analyzed and drawn using R (version 1.0.8) plotrix and Pheatmap package, respectively.

### 2.4. Statistical Analyses

The microbial diversity data, chemical composition, and fermentation quality of silage were subjected to a one-way Analysis of variance (ANOVA) using the general linear models (GLM) procedure of the Statistical Analysis System (SAS) software (version 9.3, SAS Institute Inc., Cary, NC, USA). Statistical significance (*p* < 0.05) was examined by Duncan’s test for multiple comparisons.

## 3. Results

### 3.1. Characteristics of Raw Materials before Ensiling

The chemical compositions of SCR, CS, and RS before ensiling are presented in Table 1. SCR is high in CP content (129.3 g/kg DM) and thus has a high nutritional value. However, SCR does not meet the requirements of high-quality silage for raw materials (DM content 300–350 g/kg and WSC > 50 g/kg DM) [17] because of the low DM and WSC content (166.2 g/kg and 22.9 g/kg DM). Low moisture content CS and RS were selected as the test materials in the present study and the DM content was 922.3 g/kg and 953.9 g/kg, respectively. Additionally, the DM based (g/kg DM) CP, NDF, ADF and WSC component contents of CS were 36.7, 659.0, 399.1, and 123.8, respectively. The DM based (g/kg DM) CP, NDF, ADF, and WSC component contents of RS were 34.0, 652.8, 393.0, and 146.5, respectively.

### 3.2. Nutritional Composition of CS or RS Silage Mixed with SCR

From Table 2 and Table 3, the contents of DM, CP, and WSC in each group decreased after silage. CP losses gradually decreased as the proportion of SCR in raw materials increased. As shown in Table 3, both SCR and CS mixed silage and SCR and RS mixed silage was dramatically influenced by the mixing ratio of raw material as measured by the content of DM, CP, NDF, and ADF (*p* < 0.01). In addition, the mixing ratio also had a significant effect on the WSC content in the SCR and CS mixed silage treatment groups (*p* = 0.0011). By comparing the chemical composition of SCR mixed with CS or RS, the content of DM, NDF, ADF, and WSC in the C25 group was significantly lower than that in the other two groups (*p* < 0.01), while the CP content was significantly higher than that in the other two groups (*p* < 0.01). Similarly, in the SCR and RS mixed silage treatment group, the content of DM, NDF, and ADF in the R25 group was significantly lower than that in the other two groups (*p* < 0.01), while the CP content was significantly higher than that in the other two groups (*p* < 0.01).

### 3.3. Fermentation Quality of CS or RS Silage Mixed with SCR

As shown in Table 4, the pH value was below 4.2 and the number of molds was low (<2 log10 CFU/g FM) and free of BA in all mixed silage samples. By comparing the organic acid concentration, pH, and microbial population of SCR mixed with CS or RS, the pH value, the concentration of LA, AA, and AN, and the population of LAB were dramatically influenced by the mixing ratio of raw material (*p* < 0.01). The data support the following observations, the pH value in the C25 group was significantly lower than that in the C35 group (*p* = 0.0030), and the concentration of AN was significantly lower than that in the C30 and C35 groups (*p* = 0.0016), while the LA concentration and LAB population was significantly higher than that in the C30 and C35 groups (*p* < 0.01). On the other hand, pH value and AN concentration in the R25 group were significantly lower than that in the R30 and R35 groups (*p* < 0.01), while the LA concentration was significantly higher than that in the R30 and R35 groups (*p* = 0.0010) and the LAB population was significantly higher than that in the R35 group (*p* = 0.0004). Additionally, the population of yeast was less than 2 log10 CFU/g FM in the C25 and R25 groups.

### 3.4. Microbial Community of CS or RS Silage Mixed with SCR

The bacterial community richness and diversity indices in each group of mixed silage samples are shown in Table 5. All the samples had a coverage index that reached 0.99, indicating that the identified sequences represented the majority of microbiota in silage. The results of mixed silage of SCR and CS showed that the Chao1 index first increased and then decreased as the proportion of SCR in raw materials increased, the Shannon diversity index gradually decreased, whereas the Simpson diversity index gradually increased. Group C30 had the highest Chao1 index. Group C25 had higher Simpson diversity index values and lower Shannon diversity index values. The results of mixed silage of SCR and RS showed that the Chao1 index and Simpson diversity index had an increasing trend as the proportion of SCR in raw materials increased, while the Shannon diversity index gradually decreased. From Figure 1, the number of overlapping OTUs among the seven groups of silage samples was 186 for the bacterial communities, and the OTU number of mixed SCR and CS silage was higher than that of mixed SCR and RS silage. In addition, the OTU numbers of C30 and R35 were higher than those of the other two groups under the same treatments.

Principal coordinates analysis (PCoA) was used to show the differences between the silage samples in each group according to the distance matrix of beta diversity. PCoA of bacterial communities for SCR mixed with CS or RS is shown in Figure 2; Axis 1 (18.5%) and Axis 2 (7.7%) could be interpreted as the proportion of the variance explained by the respective principal coordinate axis. The projection distance between the CS group and the RS group in Axis 1 was large, indicating that the bacterial community was clearly separated. There were some differences in the bacterial community of samples with different proportions of SCR in the CS group, while the bacterial community distances of samples with different proportions of SCR were relatively close in the RS group, indicating that they were not completely separated, and similar bacterial communities might exist.

The bacterial communities of each group were identified and 20 dominant bacteria were screened. The relative abundances of dominant bacteria in the silage samples at the genus level are shown in Figure 3. In the SCR and CS mixed silage treatment groups, *Rahnella* (41.5%) was the dominant genus in the C35 group, followed by *Lactobacillus* (39.6%), *Pantoea* (3.2%), *Serratia* (2.7%), and *Leuconostoc* (1.7%). The relative abundance of the dominant bacteria in silage samples gradually changed with the increase in the proportion of SCR in raw materials. The dominant genera in the C30 group were *Lactobacillus* (55.3%), *Rahnella* (23.8%), *Brachybacterium* (2.7%), *Serratia* (2.3%), and *Leuconostoc* (2.2%). The dominant genera in the C25 group were *Lactobacillus* (66.2%), *Rahnella* (18.4%), *Leuconostoc* (2.5%), *Lactococcus* (2.2%), and *Cupriavidus* (1.9%). In the SCR and RS mixed silage treatment group, *Lactobacillus* (65.8%), *Lactococcus* (17.6%), *Leuconostoc* (8.1%), *Cupriavidus* (2.8%), and *Weissella* (2.0%) were dominant genera in the R35 group. As the proportion of SCR in raw materials increased, the relative abundance of *Lactobacillus* increased to 77.8% in the R25 group, while the relative abundance of *Lactococcus*, *Leuconostoc*, *Cupriavidus,* and *Weissella* decreased to 13.4%, 3.8%, 1.7%, and 0.9%, respectively. From Figure 4, the relative abundance of *Janthinobacterium* of SCR was higher. In the RS treatment group, as the proportion of SCR in raw materials increased, the relative abundance of *Janthinobacterium* and *Lactobacillus* increased, while that of *Weissella* and *Leuconostoc* decreased. In the RS treatment group, the relative abundance of *Serratia*, *Pseudomonas,* and *Rahnella* decreased with increasing proportions of SCR.

## 4. Discussion

### 4.1. Characteristics of Raw Materials before Ensiling

Silage alone SCR may cause clostridial fermentation and nutrient loss and increase the risk of considerable effluent loss because of the low DM and WSC content [18]. Studies have revealed that mixing high-moisture feed with drier feedstocks, such as wheat bran, lowers moisture content and might be helpful in improving fermentation quality [19]. Furthermore, WSC plays a key role in silage fermentation processes, and a WSC content greater than 50 g/kg DM is critical for successful fermentation. As the major raw materials of ruminant feedstuff in China, CS and RS have dried sufficiently (922.3 and 953.9 g/kg for DM content, respectively), and the WSC content are 123.8 and 146.5, respectively, to support adequate fermentation. In this study, the CP content of SCR was 129.3 g/kg DM which is much higher than that of CS and RS. According to the properties of the raw materials, SCR was mixed with CS and RS in different proportions. SCR was used to improve the nutritional value, and CS and RS were added to absorb water and increase the content of WSC to improve fermentation quality.

### 4.2. Nutritional Composition of CS or RS Silage Mixed with SCR

During silage fermentation, the microorganisms in the raw material metabolize sugars to produce lactic acid, causing the content of DM and WSC to decrease. An increase in the lactic acid content results in a decrease in the pH value which inhibits the proliferation of harmful bacteria that require a large consumption of nutrients and helps in the retention of nutrients such as CP [20,21]. It has been reported that ruminants require more than 70 g/kg DM CP content in their feed to ensure normal ruminal microbial activity, and low CP content may reduce the proliferation of rumen microbes [22]. In this study, both the CP content of the C25 and R25 groups complied with the requirements (78.9 and 75.4 g/kg DM, respectively) and benefited from the abundant protein in the SCR. In addition, the SCR contained more nutrients and amino acids were abundant. Some nitrogenous substances were utilized by silage microorganisms, which synthesized the bulk of microbial protein that could be utilized by ruminants [23], thus demonstrating that the SCR can improve the nutritional value of feed. The measures of NDF and ADF were important to evaluate the nutritional value of ruminant feed. NDF and ADF are difficult to digest and absorb in feed; the acceptability of feed and animal intake decreased as NDF increased, and the digestibility of the forage decreased as ADF increased [24]. In this study, an increase in the proportion of SCR decreased the content of NDF and ADF and improved the feeding value of the mixed silage. The reason for this could be that the fiber content of the SCR was lower than that of the CS and RS.

### 4.3. Fermentation Quality of CS or RS Silage Mixed with SCR

Anaerobic environments were created by aerobic microorganisms due to the consumption of oxygen during the early stage of ensiling; aerobic microorganisms’ activity was inhibited with the growth of anaerobic microorganisms. Meanwhile, anaerobic microorganisms utilize soluble sugars to produce substantial amounts of LA under favored conditions, causing the pH to drop. In this study, the pH value was below 4.2 in all mixed silage samples and was in line with quality standards for high-quality silage.

LA concentrations and the abundance of LAB increased, while the pH value decreased as the proportion of SCR in raw materials increased. This was most likely due to the presence of multiple bioactive components such as isoflavones in SCR. As a class of important flavone compounds, soy isoflavones exist in three forms: daidzein, genistein, and glycitein [25]. In addition, soy isoflavone has better bacteriostatic effects, with two main mechanisms of action. One mechanism is to destroy the integrity of bacterial cell walls and increase the permeability of the cell membrane with the result of disrupting normal cell morphology. Another mechanism is to inhibit the growth and reproduction of bacteria by affecting DNA, RNA, and protein synthesis [26,27,28]. Isoflavones in SCR may suppress the growth and reproduction of undesirable microorganisms such as *Staphylococcus*, *Bacillus*, *Albicans*, *Listeria,* and *Enterobacter* during ensilage. The relative abundance of lactic acid bacteria increased with decreasing populations of harmful bacteria because of competition for limited nutrients, resulting in copious amounts of lactic acid being produced and a rapid decrease in pH value as well as improving the fermentation quality of mixture silages [29].

In this study, an increased proportion of SCR was accompanied by increased acetic acid content and two possible reasons that may account for this result. First, the lower levels of sugar in SCR may induce a further reduction in available sugar content during the silage process. Under sugar deficiency conditions, the silage mixture tends to transition from homofermentation to heterofermentation, producing not only lactic acid but also acetic acid, ethanol, and CO_2_ [30]. A second reason was that in the acidic environment, there was a protective mechanism wherein fermentation products transformed into compounds of weak acidity [31]. Because the acidity of acetic acid (PKa = 4.8) was weaker than that of lactic acid (PKa = 3.9), a portion of the product was converted to acetic acid. However, the protective mechanism might have limitations, it will be dampened when an external condition such as pH value is beyond this limit. This also explains why the content of acetic acid in the C25/R25 group was similar to the content of acetic acid in the C30/R30 group.

Propionic acid content progressively decreased until it disappeared as the proportion of SCR in the raw materials increased because propionate-producing mechanisms do not tolerate low pH values. Compared to lactic acid, acetic acid and propionic acid showed a stronger ability to inhibit yeast growth, so SCR might contribute to the increased aerobic stability of the silage mixture [31].

The ammonia nitrogen concentration was related to the degree of protein decomposition. An ammonia nitrogen concentration that was too high indicated the excessive decomposition of protein and may be caused by clostridial fermentation. In this study, as the proportion of SCR in raw materials increased, a rapid pH decline resulted in a lack of butyric acid, inhibited clostridial fermentation, reduced concentration of ammonia nitrogen, and reduced the consumption of protein.

### 4.4. Microbial Community of CS or RS Silage Mixed with SCR

The Alpha diversity analysis mainly included important indicators such as richness, diversity, and evenness. The Chao1 index of silage mixtures in the C30 and R25 groups was higher than that of the other silages which suggests the bacterial community richness of mixture silages in both groups. Shannon diversity gradually decreased while Simpson diversity gradually increased as the proportion of SCR in raw materials increased, suggesting that the bacterial diversity was gradually reduced. This is probably due to the fermentation quality of the silage mixture being improved by SCR; the lower pH value inhibited the growth of harmful bacteria and increased the relative abundance of lactic acid bacteria which caused a decrease in the diversity of silage microorganisms. It has been reported that the larger the relative abundance of dominant bacteria, the smaller the diversity of the microbial community will be [32], which is consistent with the findings of this study. The number of overlapping OTUs in all silage samples was 186 for the bacterial communities, indicating that even though the raw materials and ratio were different, some similar microbial communities were still involved in silage processes.

The beta diversity elucidated the change in bacterial communities in the silage mixtures. Compared with principal analysis (PCA), principal coordinates analysis (PCoA) takes the distance of the sample as a whole to consider, which was more consistent with the characteristics of the ecological data. Figure 2 shows that the bacterial communities of the two mixture silages were clearly separated which suggested that the silage raw materials had an impact on the composition of the microbial community. In addition, the silage mixture in the CS group was greatly affected by the ratio of raw materials, while the ratio of raw materials had less impact on the silage mixture in the RS group.

It was reported that the difference in microbial communities could be a critical factor in contributing to differences in silage quality [33]. During the silage process, bacteria such as *Lactobacillus*, *Weissella*, and *Lactococcus* are the main microbial species involved in acidogenic fermentation, whereas their ability to tolerate acid stress differs [10]. From Figure 3, the increase in SCR was accompanied by a drop in the pH value and an increase in the relative abundance of Lactobacillus, while the relative abundances of *Leuconostoc*, *Lactococcus,* and *Weissella* were decreased. This is attributed to the fact that these bacteria were outcompeted by acid-resistant Lactobacillus during the late phases of silage. In the silage mixtures of SCR and CS, *Rahnella* (18.4–41.5%) was the dominant bacteria in addition to Lactobacillus. *Rahnella* is a beneficial microbe in the plant rhizosphere and belongs to the Enterobacteriaceae family, which can inhibit the growth of phytopathogenic fungi and can improve the growth status of plants by nitrogen fixation [34,35]. This may be beneficial for the preservation of protein and decrease in the concentration of ammonia nitrogen.

Some studies have demonstrated that microbial metabolism would result in nutrient consumption during silage. DM is an important indicator to evaluate the nutrient contents of the silage mixture. DM content decreased with prolonged ensilage time [3] which was consistent with the findings of this study. In these mixed silages, *Lactobacillus*, *Lactococcus*, *Leuconostoc,* and *Weissella* were the dominant bacterial species with high relative abundance which decomposed the sugar substrate into compounds such as lactic acid, resulting in the decrease in WSC content. Protein breakdown was mostly caused by *Clostridium* fermentation and a relatively high plant protease activity during silage [36]. In this study, CP losses gradually decreased with the increase in the proportion of SCR in raw materials. The reason for this result could lie in the lower relative abundance of *Clostridium*. Moreover, a rapid decrease in pH with the increase in the relative abundance of *Lactobacillus*, results in a decrease in plant protease activity in the silage mixture, inhibiting the breakdown of protein.

Fermentation of *Clostridium* leads to the breakdown of protein, and *Enterobacter* and lactic acid bacteria also compete for the limited nutrient composition, so *Clostridium* and *Enterobacter* were not conducive to maintaining the fermentation quality and nutrient of mixed silage. It is worth noting that among all silage mixtures, *Clostridium* and Enterobacter were present at an exceptionally low relative abundance. This is probably due to their poor acid tolerance [37] or was possibly caused by a bacteriostatic effect of isoflavones in SCR. It was reported that *Pantoea* could reduce the concentration of ammonia nitrogen and Serratia could produce prodigiosin which inhibited the growth of fungi [32,38]. In this study, the relative abundance of *Pantoea* and *Serratia* decreased as the proportion of SCR in raw materials increased and was probably due to *Pantoea* and *Serratia’s* similarity to *Rahnella,* also belonging to *Enterobacteriaceae* families, exhibiting a somewhat lower survival rate in an acidic environment.

Interestingly, Figure 4 shows the high relative abundance of *Janthinobacterium* in SCR which creates violacein in the fermentation process. Violacein inhibits most Gram-positive bacteria such as *Bacillus subtilis* and *Staphylococcus aureus* and was inhibitory to the proliferation of fungi such as Yeast [39]. This may also be a reason why increased SCR could inhibit the growth of harmful bacteria in this study. In addition, violacein has biological activities as an antioxidant, anti-parasitic, anti-diarrheal, and immunomodulatory agent [40]. In conclusion, SCR could enhance mixture silage quality by beneficially mediating the change in the microbial community.

## 5. Conclusions

A better quality of nutrition and fermentation was shown by mixed silage with 75% SCR and 25% CS as well as 75% SCR and 25% RS. The nutrients and bioactive constituents in SCR contribute substantially to inhibiting protein degradation during silage as well as preserving the nutritional value and fermentation quality of silage mixtures by influencing the composition of the microbial community structure, increasing the relative abundance of beneficial microorganisms, and decreasing the relative abundance of undesirable microorganisms.

## Figures and Tables

**Figure 1 animals-12-00919-f001:**
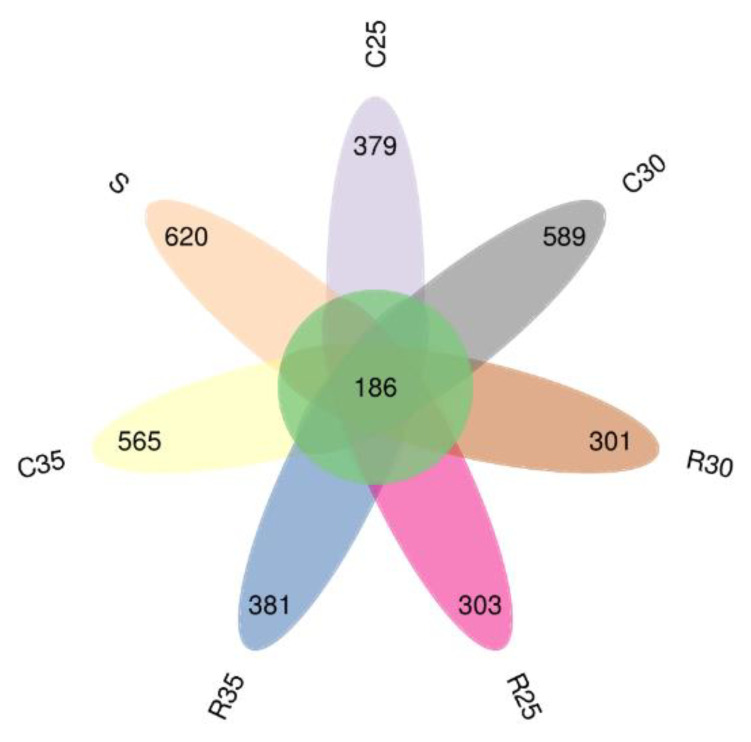
The Venn analysis of operational taxonomic units (OTUs) for corn stover or rice straw silage mixed with soybean curd residue.

**Figure 2 animals-12-00919-f002:**
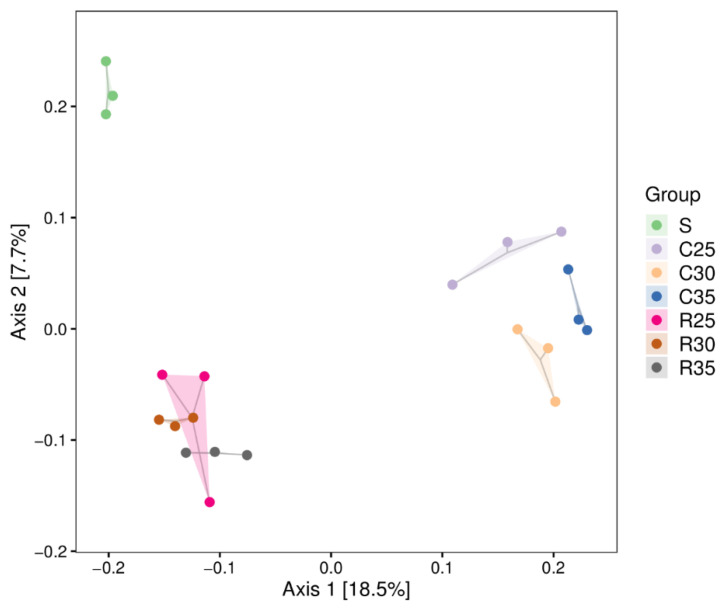
The unweighted Principal coordinate analysis (PCoA) of bacterial communities for corn stover or rice straw silage mixed with soybean curd residue (S, soybean curd residue; C35/R35, soybean curd residue: corn stover/rice straw at 65:35; C30/R30, soybean curd residue: corn stover/rice straw at 70:30; C25/R25, soybean curd residue: corn stover/rice straw at 75:25).

**Figure 3 animals-12-00919-f003:**
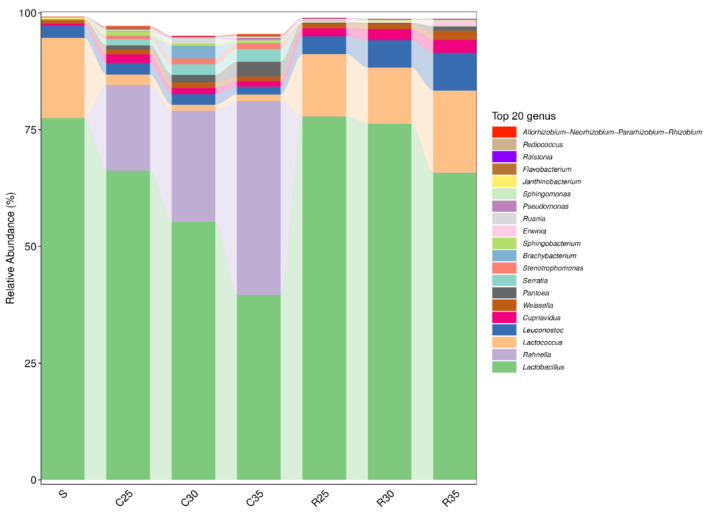
Bacterial community and relative abundance by genus for corn stover or rice straw silage mixed with soybean curd residue (S, soybean curd residue; C35/R35, soybean curd residue: corn stover/rice straw at 65:35; C30/R30, soybean curd residue: corn stover/rice straw at 70:30; C25/R25, soybean curd residue: corn stover/rice straw at 75:25).

**Figure 4 animals-12-00919-f004:**
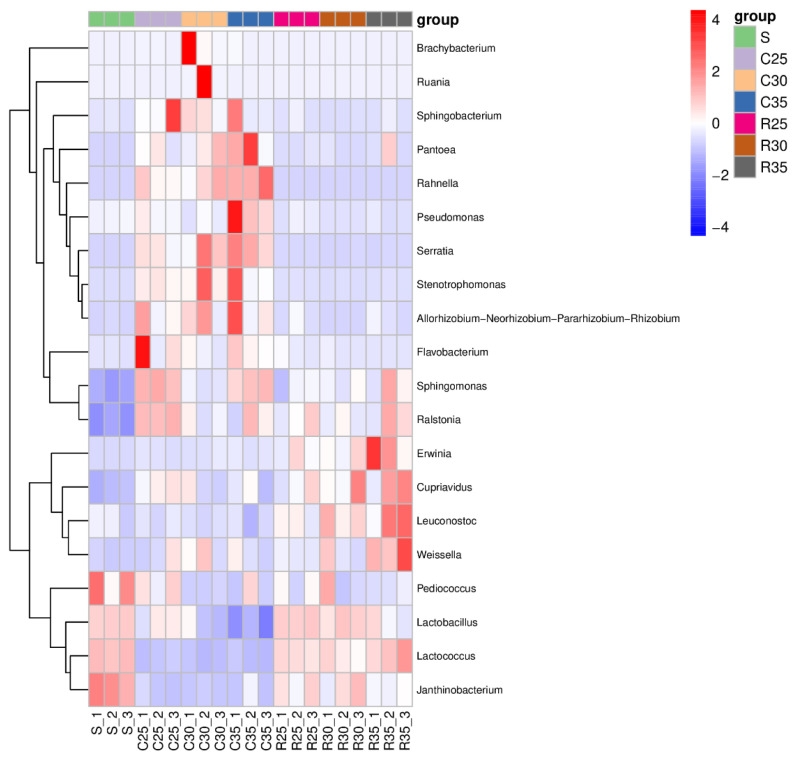
Heatmap analysis of the main bacterial community of corn stover or rice straw silage mixed with soybean curd residue (S, soybean curd residue; C35/R35, soybean curd residue: corn stover/rice straw at 65:35; C30/R30, soybean curd residue:corn stover/rice straw at 70:30; C25/R25, soybean curd residue: corn stover/rice straw at 75:25).

**Table 1 animals-12-00919-t001:** Chemical composition of soybean curd residue, corn stover, and rice straw (Mean ± SD, *n* = 3).

Items ^1^	Soybean Curd Residue	Corn Stover	Rice Straw
DM, g/kg FM	166.2 ± 3.21	922.3 ± 8.45	953.9 ± 5.82
CP, g/kg DM	129.3 ± 0.31	36.7 ± 0.65	34.0 ± 0.94
NDF, g/kg DM	552.9 ± 8.01	659.0 ± 5.49	652.8 ± 5.77
ADF, g/kg DM	300.5 ± 4.26	399.1 ± 5.60	393.0 ± 3.67
WSC, g/kg DM	22.9 ± 0.13	123.8 ± 5.24	146.5 ± 4.65

^1^ FM, fresh matter; DM, dry matter; CP, crude protein; NDF, neutral detergent fiber; ADF, acid detergent fiber; WSC, water soluble carbohydrate.

**Table 2 animals-12-00919-t002:** Chemical composition of each group before mixed silage (Mean ± SD, *n* = 3).

Items ^1^	C35	C30	C25	R35	R30	R25
DM, g/kg FM	431.2 ± 6.15	385.7 ± 3.76	349.1 ± 3.74	437.1 ± 3.05	402.9 ± 1.39	361.5 ± 4.42
CP, g/kg DM	96.2 ± 1.24	101.5 ± 0.38	107.1 ± 0.35	95.4 ± 0.63	101.3 ± 0.86	106.5 ± 0.55
NDF, g/kg DM	595.4 ± 1.55	584.8 ± 1.16	578.3 ± 6.77	586.7 ± 2.94	580.2 ± 3.06	574.5 ± 1.49
ADF, g/kg DM	337.2 ± 3.72	330.2 ± 1.54	322.3 ± 1.16	334.9 ± 2.88	326.8 ± 3.89	320.4 ± 2.31
WSC, g/kg DM	58.1 ± 0.51	53.3 ± 1.21	50.8 ± 0.37	67.7 ± 1.59	59.0 ± 1.12	53.7 ± 0.79

^1^ FM, fresh matter; DM, dry matter; CP, crude protein; NDF, neutral detergent fiber; ADF, acid detergent fiber; WSC, water soluble carbohydrate; C35 (R35), soybean curd residue: corn stover (rice straw) at 65:35; C30 (R30), soybean curd residue: corn stover (rice straw) at 70:30; C25 (R25), soybean curd residue: corn stover (rice straw) at 75:25.

**Table 3 animals-12-00919-t003:** Chemical composition of soybean curd residue mixed with corn stover or rice straw.

Items ^1^	Corn Stover					Rice Straw				
	C35	C30	C25	SEM	*p* Value	R35	R30	R25	SEM	*p* Value
DM (g/kg FM)	421.0 ^a^	380.1 ^b^	347.9 ^c^	1.87	<0.0001	430.1 ^a^	394.9 ^b^	354.2 ^c^	2.26	<0.0001
CP (g/kg DM)	34.0 ^c^	55.6 ^b^	78.9 ^a^	1.27	<0.0001	29.6 ^c^	52.0 ^b^	75.4 ^a^	0.73	<0.0001
NDF (g/kg DM)	606.3 ^a^	581.4 ^b^	568.3 ^c^	4.36	0.0023	629.7 ^a^	609.3 ^b^	586.0 ^c^	4.78	0.0020
ADF (g/kg DM)	377.0 ^a^	357.6 ^b^	342.3 ^c^	2.27	0.0001	389.3 ^a^	370.7 ^b^	363.6 ^c^	3.01	0.0026
WSC (g/kg DM)	36.5 ^a^	32.3 ^b^	28.6 ^c^	0.77	0.0011	39.2 ^a^	34.7 ^ab^	31.8 ^c^	1.69	0.0560

^1^ FM, fresh matter; DM, dry matter; CP, crude protein; NDF, neutral detergent fiber; ADF, acid detergent fiber; WSC, water soluble carbohydrate; SEM, standard error of means; C35 (R35), soybean curd residue: corn stover (rice straw) at 65:35; C30 (R30), soybean curd residue: corn stover (rice straw) at 70:30; C25 (R25), soybean curd residue: corn stover (rice straw) at 75:25. ^a–c^ Means in the same column followed by different letters differ significantly (*p* < 0.05).

**Table 4 animals-12-00919-t004:** Organic acids concentration, pH, and microbial population of soybean curd residue mixed with corn stover or rice straw.

Items ^1^	Corn Stover					Rice Straw				
	C35	C30	C25	SEM	*p* Value	R35	R30	R25	SEM	*p* Value
pH	4.01 ^a^	3.93 ^b^	3.86 ^b^	0.02	0.0030	3.96 ^a^	3.88 ^b^	3.85 ^c^	0.02	0.0019
AN (g/kg TN)	50.8 ^a^	47.8 ^b^	45.1 ^c^	0.59	0.0016	47.5 ^a^	43.6 ^b^	39.9 ^c^	0.67	0.0007
LA (g/kg DM)	15.8 ^c^	18.2 ^b^	21.5 ^a^	0.51	0.0007	11.5 ^c^	14.1 ^b^	16.4 ^a^	0.50	0.0010
AA (g/kg DM)	21.4 ^b^	27.3 ^a^	27.1 ^a^	0.58	0.0006	30.2 ^b^	43.3 ^a^	42.5 ^a^	0.63	<0.0001
PA (g/kg DM)	5.37	2.61	ND	-	-	4.17	ND	ND	-	-
BA (g/kg DM)	ND	ND	ND	-	-	ND	ND	ND	-	-
LAB (log10 CFU/g FM)	7.11 ^c^	7.87 ^b^	8.25 ^a^	0.04	<0.0001	7.33 ^b^	7.85 ^a^	7.96 ^a^	0.06	0.0004
Yeast (log10 CFU/g FM)	2.82	2.48	<2.00	-	-	2.84	2.01	<2.00	-	-
Mold (log10 CFU/g FM)	<2.00	<2.00	<2.00	-	-	<2.00	<2.00	<2.00	-	-

^1^ FM, fresh matter; DM, dry matter; AN, ammonia-N; TN, total N; LA, lactic acid; AA, acetic acid; PA, propionic acid; BA, butyric acid; LAB, lactic acid bacteria; CFU, colony forming units; ND, not detected; SEM, standard error of means. ^a–c^ Means in the same column followed by different letters differ significantly (*p* < 0.05).

**Table 5 animals-12-00919-t005:** Alpha diversity of the bacterial community of corn stover or rice straw silage mixed with soybean curd residue.

Item	Corn Stover			Rice Straw		
	C35	C30	C25	R35	R30	R25
Chao1	833.64	901.73	775.93	657.68	707.66	710.39
Shannon	5.91	5.80	5.77	5.59	5.52	5.48
Simpson	0.918	0.948	0.954	0.936	0.943	0.949
Coverage	0.997	0.996	0.997	0.997	0.997	0.997

## Data Availability

The data presented in this study are available in article.

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
