# Peer review of "Fermentation Quality and Microbial Community of Corn Stover or Rice Straw Silage Mixed with Soybean Curd Residue"

_animals, 2022, doi:10.3390/ani12070919_

Round 1

Reviewer 1 Report

Dear Authors,

The objective of the study was to evaluate the chemical constituents, fermentation quality, and microbial community of corn stover (CS) or rice straw (RS) mixed with soybean curd residue (SCR) silages. Additionally, It was tested three ratios of SCR and CS or RS (75:25, 70:30, and 65:35). The paper is well written, concise, and clear. The study itself appeared well designed and conducted, and appears to have addressed the specific objectives stated.   

Some minor comments are listed below that the authors might consider:

L13 - Please rewrite the sentence. It is not clear. What did you mean with "preservation problem"?

L23 - Various parameters? Please list some parameters.

L73-76 - Please rewrite the objective of the study. Please consider rewriting the objective like is being presented in the abstract section.

L139-143 - Please include all statistical procedures adopted.

L290 - Please change palatability to acceptability.

L295 - 393 - PLease reorganize the information, and divide in five or 6 paragraphs per subtopic;

L395-399 - Please mention the proportions included in the study.

Reviewer 2 Report

This manuscript is very interesting and well written. In my opinion, it should be admitted to publication, however, it requires a small adjustment and supplementation of information:

  • I suggest more specifically describe the aim of research.
  • In my opinion, the authors must justify the choice of combinations of blends: 75:25 (C25/R25), 70:30 (C30/R30), and 65:35(C35/R35).
  • Principal coordinate analysis explain only 18.5% and 7.7% of bacterial communities (see Figure 2). Are these values sufficient to explain observed dependencies? I am asking for a comment in the text in this regard.
  • Figure 3 – In my opinion, the results obtained must be correlated with the results of chemical analyzes of studied mixtures, because the data tables 1 and 2 show that these results are very important and decisive.
  • Conclusions – In my opinion, they should correspond to the aim of research. The types of mixtures tested in this work should be taken into account, with the indication of the best mix.
